# Mechanistic Studies and a Retrospective Cohort Study: The Interaction between PPAR Agonists and Immunomodulatory Agents in Multiple Myeloma

**DOI:** 10.3390/cancers14215272

**Published:** 2022-10-27

**Authors:** Jian Wu, Emily Chu, Barry Paul, Yubin Kang

**Affiliations:** Division of Hematologic Malignancies and Cellular Therapy, Department of Medicine, Duke University Medical Center, Durham, NC 27710, USA

**Keywords:** immunomodulatory drugs, PPAR, diabetes, dyslipidemia, CpG island, methylation, metabolomics, treatment response, survival

## Abstract

**Simple Summary:**

Cereblon (CRBN) is a direct binding target of immunomodulatory drugs (IMiDs) that are commonly used to treat multiple myeloma (MM). Many patients with MM have comorbidities, including diabetes and/or dyslipidemia, and are treated with peroxisome proliferator-activated receptor (PPAR) agonists. This study aimed to further analyze the effects and mechanisms underlying the drug-to-drug interactions between IMiDs and PPAR agonists in MM. We found that PPAR agonists reduced CRBN expression by inducing DNA methylation and increasing protein degradation. PPAR agonists and IMiDs showed opposing metabolic effects in MM cells. Our retrospective study suggested an inferior response and outcome when PPARs and IMiDs were concurrently administered. Our study has important implications for the care of patients with MM and provides a foundation for exploring novel compounds or PPAR partial agonists/antagonists that can increase CRBN expression while retaining their lipid-lowering or insulin-sensitizing functions.

**Abstract:**

Our previous study demonstrated that peroxisome proliferator-activated receptor (PPAR) agonists downregulated cereblon (CRBN) expression and reduced the anti-myeloma activity of lenalidomide in vitro and in vivo. We aimed to determine whether DNA methylation and protein degradation contribute to the effects of PPAR agonists. CRBN promoter methylation status was detected using methylation-specific polymerase chain reaction. The CRBN protein degradation rate was measured using a cycloheximide chase assay. Metabolomic analysis was performed in multiple myeloma (MM) cells treated with PPAR agonists and/or lenalidomide. Our retrospective study determined the effect of co-administration of PPAR agonists with immunomodulatory drugs on the outcomes of patients with MM. CpG islands of the CRBN promoter region became highly methylated upon treatment with PPAR agonists, whereas treatment with PPAR antagonists resulted in unmethylation. The CRBN protein was rapidly degraded after treatment with PPAR agonists. Lenalidomide and fenofibrate showed opposite effects on acylcarnitines and amino acids. Co-administration of immunomodulatory drugs and PPAR agonists was associated with inferior treatment responses and poor survival. Our study provides the first evidence that PPAR agonists reduce CRBN expression through various mechanisms including inducing methylation of CRBN promoter CpG island, enhancing CRBN protein degradation, and affecting metabolomics of MM cells.

## 1. Introduction

Multiple myeloma (MM) is an incurable cancer of terminally differentiated plasma cells and is the second most common hematological malignancy in the Western world [1]. Immunomodulatory drugs (IMiDs), including lenalidomide, pomalidomide, and thalidomide, are highly effective treatments for patients with MM [2]. Although the combination of IMiDs with other drugs such as proteasome inhibitors and antibodies (e.g., daratumumab, elotuzumab, and isatuximab) can induce remission in most patients, almost all patients eventually relapse because the MM cells acquire resistance to one or several of the drugs [3].

MM is more common in elderly individuals; patients in this age group often suffer from other comorbidities such as diabetes and dyslipidemia [4,5,6]. Over 1 in 4 patients with MM has diabetes and 29.3% of myeloma patients have dyslipidemia [7]. Peroxisome proliferator-activated receptor (PPAR) agonists are U.S. Food and Drug Administration (FDA)-approved drugs indicated for use in type 2 diabetes and dyslipidemia [8,9]. Both the comorbidities and their treatment can have a severe impact on patients or can severely interact with the natural biology of MM [4].

PPARs belong to the nuclear receptor family and comprise an N-terminal domain, a DNA binding domain, a hinge region, and a ligand binding domain [10]. Three different PPAR subtypes have been identified (PPARα, PPARβ/δ, and PPARγ). Each shows a distinct expression pattern but can also be co-expressed in numerous cell types [11,12,13]. Upon activation with a high-fat diet, ligands, or agonists, PPARs heterodimerize with the 9-cis-retinoic acid receptor and bind to PPAR response elements (PPREs) that consist of a direct repeat of two conserved AGGTCA hexamers separated by a single base that are referred to as DR1 elements [14,15,16]. PPARs regulate lipid metabolism, insulin sensitivity, and adipocyte differentiation [11].

Cereblon (CRBN) is a substrate adaptor in the CRL4^CRBN^ E3 ubiquitin ligase complex, that alters the substrate specificity of the complex following binding with IMiDs [17,18,19]. The binding of IMiDs to CRBN results in the ubiquitination and degradation of several factors, including the IKZF1 and IKZF3 transcription factors [20,21]. The degradation of these factors inhibits the survival and progression of MM cells [22,23,24]. Our previous data demonstrated that PPARs (PPARα, PPARβ/δ, and PPARγ) negatively regulate CRBN transcription activity through binding to the CRBN promoter region [25]. Moreover, co-treatment with PPAR agonists decreased the anti-myeloma activity of lenalidomide in vitro and in vivo. However, further studies are required to clarify the exact molecular mechanisms involved in this process, particularly epigenetic regulation. In addition, whether PPAR agonists affect CRBN protein degradation needs to be determined.

In this study, we conducted in-depth research to uncover the underlying mechanisms of epigenetic regulation, specifically DNA methylation. Additionally, a large-scale retrospective cohort study was performed to determine the effect of co-administration of PPAR agonists on the outcomes of patients with MM treated with IMiDs. To our knowledge, our study is the first to investigate the underlying mechanisms of this drug-to-drug interaction. Our findings provide a new avenue to investigate the effects of PPAR agonists that are commonly used for the treatment of diabetes and dyslipidemia on IMiD activity and to develop novel agents targeting the PPAR pathway for enhanced anti-myeloma activity.

## 2. Materials and Methods

### 2.1. Ethics Approval

This is a retrospective study that did not involve patient intervention or the need for obtaining clinical specimens, and all the data were analyzed anonymously. Therefore, there is no need for informed consent, also no need an institutional review board approve.

### 2.2. Cell Lines

MM cell lines used in this study included MM1R, MM1S, RPMI8226, NCIH929, and U266. MM1.S (ATCC CRL-2974) and MM1.R (ATCC CRL-2975) cells were purchased from ATCC (Rockville, MD, USA). RPMI8226 (631-CCL-155), NCIH929 (540-CRL-9068), and U266 (TIB-196) were purchased from Duke Cell Culture Facility (CCF). All cell lines were cultured at 37 °C under 5% CO_2_ in RPMI1640 medium supplemented with 2 mM Glutamax and 10% fetal calf serum (Mediatech, Herndon, VA, USA).

### 2.3. Methylation Specific PCR (MSP)

The genomic DNA from cultured cells was extracted and subjected to methylation specific-PCR analysis as described [26]. Briefly, MM cells were treated with DMSO control, fenofibrate (PPARα agonist, 5 μM), GW5015016 (PPARβ/δ agonist, 5 μM), troglitazone (PPARγ agonist, 5 μM), GW6471 (PPARα antagonist, 10 μM), GSK3787 (PPARβ/δ antagonist, 10 μM), or GW9662 (PPARγ antagonist, 10 μM) for 48 h and cells were harvested. Tissue/Cell genomic DNA isolation kit with the Wizard DNA clean-up stream (Promega, Madison, WI, USA) was used to isolate DNA. The genomic DNA was then treated with bisulfite per the manufacture’s instruction (EZ DNA Methylation Kit, Zymo Research, Irvine, CA, USA). PCR was performed using two CRBN promoter specific primers recognizing the methylated and unmethylated CpG sites. Primers were designed to detect the methylation status of CpG sites using MethPrimer program (http://www.urogene.org/methprimer/, accessed on 1 February 2022). The primer pair for the methylated form (129 bp) was as follows: Forward: GAATAAAGTGAGGGTTTTGTAGC; Reverse: ACCTAAAAATAATAACCTAAACGAA. The primer pair for the unmethylated form (131 bp) was as follows: Forward: TGGAATAAAGTGAGGGTTTTGTAGT; Reverse: ACCTAAAAATAATAACCTAAACAAA. PCR amplification was performed as follows: 95 °C for 5 min; 40 °Cycles of 95 °C for 45 s, 60 °C for 45 s, 72 °C for 45 s; and finally, 10 min at 72 °C. The PCR products were separated by electrophoresis in 2% agarose gels and visualized after staining with ethidium bromide.

### 2.4. Protein Degradation Analysis (CHX Chase)

Protein degradation assays were based on the use of protein synthesis inhibitor cycloheximide (CHX). MM1.R and NCIH929 cells were treated with 5 µM PPAR agonist or DMSO for 3 h and then CHX (100 µg/mL) was added. Cells were collected at various time point (0, 2, 4, 6, 8 and 10 h) after CHX treatment, and whole cell lysate was prepared and analyzed for CRBN expression by Western blot analysis and the signal density was quantitated by Image J [27,28].

### 2.5. Reagents and Antibodies

PPARα antibody was obtained from Santa Cruz (Cat#: SC-398394) and Abcam (Cat#: ab227074). PPAR-β/δ antibody was purchased from Novus Biologicals LLC (Cat#: NBP2-22468) and ABclonal (Cat#: A5656). PPARγ antibody was purchased from Novus Biologicals LLC (Cat#: NBP2-22106) and Thermo Fisher (Cat#: 16643-1-AP). IKZF1 antibody was purchased from Santa Cruz Biotechnology, Inc. (Dallas, TX, USA) and Novus Biologicals (Cat#: NBP1-98314). IKZF3 antibody was purchased from Novus (Cat#: NBP2-46048). CRBN antibody was obtained from Sigma-Aldrich (St. Louis, MO, USA). EZH2 antibody (Cat#4905S) and H3K27me3 antibody (Cat #9733T) were purchased form Cell Signaling Technology.

### 2.6. Western Blot Analysis

MM cells were harvested, washed with PBS, and re-suspended in lysis buffer containing 50 mM Tris-HCl pH 7.4, 150 mM NaCl, 1 mM EDTA, 1% Triton ×100, 1% Sodium deoxycholate, and 0.1% SDS. The cells were further lysed by brief sonication. The lysates were centrifuged at high speed for 15 min to remove the cell debris. Total protein was quantified using Dc protein estimation kit (Bio Rad) with BSA for standard curve. Approximately 20 μg protein was loaded and run on SDS PAGE. The proteins were transferred onto nitrocellulose membrane. The membrane was blocked with 5% BSA in Tris-Buffered Saline containing 0.05% of Tween 20 (TBST) and then incubated with indicated primary antibodies at 4 °C overnight. The membrane was then probed with HRP-conjugated secondary antibody and developed using Pierce ECL substrate.

### 2.7. Metabolomics

MM cells were treated with DMSO, fenofibrate, lenalidomide or the combination of fenofibrate and lenalidomide for 48 h. The cells were washed three times with ice-cold PBS and resuspended in 160 µL of ice-cold deionized water containing 0.6% formic acid. The cell suspension was then added 150 µL acetonitrile, mixed well with vortex, and stored at −80 °C until metabolomics analysis. Targeted metabolomics assays for acylcarnitines and amino acids were performed at Duke Metabolomics Core Facility using tandem mass spectrometry. The data were acquired using a Waters triple quadrupole detector equipped with Acquity UPLC system and controlled by Mass Lynx 4.1 software platform (Waters, Milford, MA, USA) [29].

### 2.8. Patient Population and Studies

The retrospective cohort study was to compare the treatment response, progression free survival (PFS), and overall survival (OS) between myeloma patients with co-existing type II diabetes and/or dyslipidemia managed with a PPAR agonist concurrently with an IMiD containing therapy for their myeloma and myeloma patients with type II diabetes and/or dyslipidemia who did not receive a PPAR agonist at any point but treated with other diabetes agents (metformin, insulin etc.) or lipid-lowering agents (Statins) during the course of their anti-myeloma treatment. Duke medical records were queried using parameters that included: (a) myeloma patients (ICD9 203.* or ICD10 C90.*) with Type II diabetes and/or hyperlipidemia (ICD9 250.*, 249.*, 272.* or ICD10 E08.*, E09.*, E10.*, E11.*, E13.*, E78.*) diagnosed between 2014 and 2020; (b) prescribed (at Duke) an IMiD (thalidomide, lenalidomide or pomalidomide) between 2014 and 2020; and with or without (c): prescribed or dispensed or administered an PPAR agonist, including fenofibrate, gemfibrozil, cirpofibrate, bezafibrate, pioglitazone, rosiglitazone, or troglitazone, between 2014 and 2020. Clinical characteristics of these patients were summarized in Appendix A.

Response to treatment was defined according to the International Myeloma working group (IMWG) treatment response criteria and classified as complete remission (CR), very good partial response (VGPR), partial response (PR), stable disease (SD), or progressive disease (PD) [30]. Cytogenetic risk stratification was performed based on the IMWG criteria and disease was staged according to the International Stage system (ISS) [31]. PFS was defined as the interval from the date of diagnosis to the last follow-up date or the date of disease recurrence. OS was defined as the duration between the date of treatment initiation to the date of death or last follow-up, with those alive censored on the date of last contact.

### 2.9. Statistical Analysis

Summary of patient characteristics as well as patient follow-up were tabulated. Kaplan–Meier estimation was used to determine median OS, as well as PFS, stratified by the presence of the PPAR agonist. In addition, multivariate Cox regression analysis was performed using the stepwise Cox regression model to assess the effect of multiple independent prognostic factors on survival outcome. The Log-Rank test was used to test for differences in OS and PFS among patients with or without PPAR agonists. Chi test and logistic regression analysis were used to study variables predictive of response.

Values reported and shown in graphical displays are the mean ± standard error of the mean, as indicated. Comparisons of mean expression across groups were made using two-sample *t*-tests. It was decided to use a *t*-test with equal or unequal variances based on the distribution within groups. For all comparisons, *p* < 0.05 was used to denote significance. 

## 3. Results

### 3.1. High DNA Methylation at CpG Islands of the CRBN Promoter Region Was Associated with Acquisition of Resistance to Lenalidomide in MM Cell Lines

DNA methylation is a biological process by which methyl groups are added to DNA molecules. DNA methylation in a gene promoter typically represses gene transcription [32]. In our previous research, we checked the sensitivity to lenalidomide in multiple MM cell lines and found that NCIH929 and RPMI8226 (IC50 of 15.39 μM and 17.55 μM, respectively) were relatively sensitive to lenalidomide while U266, OPM1, OPM2, MM1.R, INA6 were resistant to lenalidomide (IC50 of 58.87 μM, 33.71 μM, 42.78 μM, 34.36 μM, and 38.29 μM) [25]. To evaluate whether DNA hypermethylation correlates with resistance to lenalidomide, the DNA methylation status of CpG islands in the CRBN promoter region was evaluated in NCIH929 and RPMI8226 cells (lenalidomide-sensitive MM cell lines) as well as in MM1.R, MM1.S, and U266 cells (lenalidomide-resistant MM cell lines). We utilized the MethPrimer program [33] to predict CpG islands in the promoter region of CRBN (Figure 1A). One CpG island was found from −1724 to −2014 bp and contained 10 CpG sites (Figure 1B). Methylated and unmethylated DNAs in the CpG island of the CRBN promoter region were measured using methylation-specific polymerase chain reaction (MSP). MM1.R, MM1.S, and U266 cells were hypermethylated in CpG islands, while NCIH929 and RPMI8226 cells were partially methylated (Figure 1C). This result is consistent with our previous results, which showed higher CRBN expression levels in NCIH929 and RPMI8226 cells and lower CRBN expression levels in MM1.R, MM1.S, and U266 cells. These data suggest that DNA methylation of CpG islands in the CRBN promoter region may be associated with lenalidomide resistance in MM cell lines.

### 3.2. Exposure to PPAR Agonists Induced DNA Methylation in the CpG Islands of the CRBN Promoter Region in MM Cell Lines

Our previous study showed that PPARs bind to the CRBN promoter region and directly repress CRBN transcription in MM cells [25]. Additionally, we demonstrated that fenofibrate (PPARα agonist), GW5015016 (PPARβ/δ agonist), and troglitazone (PPARγ agonist) enhance PPAR binding to the CRBN promoter and reduce CRBN expression in the mRNA and protein levels. In contrast, PPAR antagonists (GW6471, GSK3787, and GW9662) increased CRBN expression and augmented the anti-myeloma activity of lenalidomide.

When PPAR binds to PPREs in its target region, it recruits positive and negative coregulatory proteins (coactivators and corepressors, respectively) that covalently modify histone proteins, resulting in acetylation/deacetylation and methylation/demethylation [11,34,35,36,37,38]. DNA methylation is a vital epigenetic modification that is involved in the regulation of numerous biological processes including embryogenesis, transcriptional regulation, and cell differentiation [39,40,41]. To determine the epigenetic mechanisms, we performed MSP, which would evaluate the methylation pattern of CpG islands in the CRBN promoter region upon exposure to PPAR agonists in MM cell lines, including the lenalidomide-sensitive and -resistant MM cell lines. We found that PPAR agonists induced methylation in the CpG island of the CRBN promoter region. We detected complete methylation in the CpG islands of the CRBN promoter region in the PPAR agonist-treated group, while only partial methylation was observed in the DMSO group. Similar results were observed across all tested MM cell lines (Figure 2A).

In contrast, treatment with PPAR antagonists resulted in demethylation of the CpG island of the CRBN promoter region. We treated MM1.R, NCIH929, and RPMI8226 cells with PPAR antagonists (GW3787, GW9662, and GW6471). MSP results showed that PPAR antagonists reversed the methylation pattern from being methylated to unmethylated (Figure 2B). These data demonstrate an important role of PPARs in regulating the methylation/demethylation of the CRBN promoter region.

Enhancer of zeste homolog 2 (EZH2), a histone-lysine N methyltransferase, catalyzes the addition of methyl groups to histone H3 at lysine 27, thereby repressing gene transcription and function [42]. Thus, we determined the effects of PPAR agonists and antagonists on EZH2 and H3K27me. MM1.R and NCIH929 cells were treated with PPAR agonists and antagonists for 48 h. Total protein was lysed for Western blotting. We found that PPAR agonists (fenofibrate, GW501516, and troglitazone) increased the levels of EZH2 and H3K27me, while PPAR antagonists (GW3787, GW9662, and GW6471) decreased their expression levels (Figure 2C, Appendix A). These results suggest that PPAR regulates the CRBN methylation status through the EZH2/H3K27me pathway, thereby affecting CRBN expression.

### 3.3. CRBN Is Rapidly Degraded in the Presence of PPAR Agonists

Proteasomal degradation is a key post-translational mechanism regulating protein levels and biological functions. In addition to serving as a transcription factor, PPARγ is an E3 ubiquitin ligase that causes ubiquitination and proteolytic degradation of NF-κB [43] and MUC1-C [44]. To evaluate whether CRBN is regulated by PPAR agonists via this mechanism, a cycloheximide degradation assay was performed, which would detect the degradation rate of CRBN in the presence or absence of PPARs agonists. Our previous data suggested that 5 µM of PPAR agonists effectively suppress CRBN expression in the mRNA and protein levels. Here, we treated MM1.R and NCIH929 cells with 5 µM of PPAR agonists, followed by cycloheximide and then measured CRBN protein level at different time points. We observed significant enhancement in CRBN protein degradation at 2–4 h with PPAR agonist treatment (Figure 3A and Appendix A). Furthermore, we tested whether PPAR agonist treatment promotes CRBN degradation in a proteasome-dependent manner. To this end, MM1.R and NCIH929 cells were treated with only PPAR agonist, only proteasome inhibitor (bortezomib), or a combination of PPAR agonist and bortezomib. Bortezomib partially but reproducibly protected against CRBN degradation induced by PPAR agonist treatment (Figure 3B and Appendix A). These data suggest that PPAR agonists, at least in part, increase CRBN proteasomal degradation.

### 3.4. Fenofibrate and Lenalidomide Have Opposite Effects on Lipid and Amino Acid Metabolism

PPARs play an important role in lipid oxidation by upregulating the gene expression of acyl-CoA oxidase and carnitine palmitoyl transferase I, which are enzymes involved in β-oxidation [45,46,47,48]. Fenofibrate, a PPARα agonist, is widely prescribed for the treatment of dyslipidemia, type 2 diabetes, and metabolic syndrome. Fenofibrate repairs liver injury by decreasing acylcarnitine levels and activating fatty acid β-oxidation genes that cause intracellular oxidative stress (e.g., Cpt1b, Cpt2, Mcad, and Hadha) [49]. However, the adverse effect of fenofibrate of decreasing amino acids by stimulating their catabolism has also been observed [49,50].

To further explore the mechanism of drug-to-drug interactions between PPAR agonists and IMiDs, a comprehensive metabolic analysis was performed using a combination of fenofibrate and lenalidomide, which would identify metabolite differences between the combination and control groups. Our data suggested that treatment with fenofibrate reduces the levels of C2 and C5 acylcarnitines (Figure 4A), long- and medium-chain acylcarnitines (Figure 4B,C), and several amino acids (Figure 4D), while the combination treatment attenuates the metabolic modulatory effect of fenofibrate. The lenalidomide treatment group showed increased acylcarnitine and amino acid levels, which was the opposite to fenofibrate’s effect. This result provides new insights into the molecular mechanism of the drug-to-drug interactions and could be relevant both for their therapeutic action and metabolic adverse effects.

### 3.5. Retrospective Cohort Study: Co-Administration of PPAR Agonists with IMiDs Results in Worse PFS and OS in Patients with MM

We performed a retrospective cohort study to investigate the impact of co-administration of PPAR agonists with IMiDs on the clinical outcomes of patients with MM. A total of 196 patients with MM were included in the study—of these patients, 114 had MM with co-existing type II diabetes and/or dyslipidemia and received IMiDs but not a PPAR agonist during the course of their anti-myeloma treatment (they received other diabetes medications, such as metformin or insulin, and/or statins for dyslipidemia); 82 patients had MM with co-existing type II diabetes and/or dyslipidemia and received a PPAR agonist concurrently with an IMiD-containing therapy for MM management. Demographic and clinical features at baseline are shown in Appendix A. The median ages at diagnosis for the IMiD group and IMiD with PPAR agonist group were 64.5 years (range 40–88) and 64.7 years (range 38–85), respectively. There were no significant differences in age, sex, race, body mass index (BMI), International Staging System (ISS) stage, or cytogenetic risk between the two groups of patients. 54.4% and 54.9% of the IMiD and IMiD/PPAR agonist cohorts, respectively, received hematopoietic stem cell transplantation treatment (HSCT). The PPAR agonists comprised known antidiabetic drugs including gemfibrozil, fenofibrate, and pioglitazone.

Co-administration of PPAR agonists and IMiDs was associated with a lower complete response (CR) rate, lower very good partial response (VGPR) rate, and lower overall response rate (Appendix A). The IMiD group had a 24.6% CR rate and 35.1% VGPR rate. In contrast, the IMiD/PPAR cohort had only an 11% CR rate and 26.82% VGPR rate. The overall response rate was 90.4% in the IMiD-alone group and 64.6% in the IMiD/PPAR agonist group (*p* < 0.0001). Additionally, progression free survival (PFS) was significantly reduced in patients with MM taking concurrent PPAR agonists (median: 21.3 months) than in similar patients taking no PPAR agonists (median: 37.9 months) (*p* = 0.0374) (Figure 5, left panel). Furthermore, overall survival (OS) was significantly reduced in patients taking concurrent PPAR agonists (median: 70.2 months) than in patients not taking PPAR agonists (median: 114 months) (*p* = 0.0416) (Figure 5, right panel).

In addition, multivariate Cox regression analysis further suggested that this combination strategy was an independent prognostic risk factor for poor outcomes (*p* = 0.029, HR = 0.559, 95% CI: 0.332–0.941). Age, body mass index, and HSCT treatment were also significant independent factors affecting MM outcomes (*p* = 0.033, HR = 0.501, 95% CI: 0.265–0.946; *p* = 0.043, HR = 1.382, 95% CI: 1.010–1.893; *p* < 0.0001, HR = 0.325, 95% CI: 0.189–0.559, respectively) (Appendix A). These data further demonstrate that the combination of PPAR agonists and IMiDs is significantly associated with an unfavorable OS.

## 4. Discussion

High CRBN expression in patients with MM is associated with a good clinical response to IMiDs, while inactivation of the CRBN gene in cell lines and low protein expression confer resistance to IMiD treatment [51,52]. Therefore, CRBN levels are expected to be a promising marker for the efficacy of IMiDs. However, it remains unclear how CRBN is regulated and whether different genomic approaches or drugs could affect sensitivity to IMiDs through the regulation of CRBN expression. We have previously shown that PPARs agonists (fenofibrate, GW501516, and troglitazone) reduce the anti-myeloma effect of lenalidomide by downregulating the transcription activity of CRBN, while PPARs antagonists (GW3787, GW9662, and GW6471) increase CRBN expression and improve lenalidomide activity [53]. Promoter binding analysis based on the Chip assay and luciferase assay suggested that there are different PPAR binding sites on the CRBN promoter region; therefore, PPARs regulate CRBN transcriptional activity by stimulating these sites [53]. In this study, we explored the exact mechanism of this transcriptional inhibition of CRBN by PPARs. We observed hypermethylation of CpG islands in the CRBN promoter in lenalidomide-resistant MM cell lines, while partial methylation was observed in lenalidomide-sensitive MM cell lines. These data are consistent with those of our previous study [25]. Importantly, our MSP analyses demonstrated that PPAR agonists affect the CRBN promoter CpG island methylation patterns and cause hypermethylation in CRBN promoter region. At the same time, PPAR antagonists changed the patterns from methylated to unmethylated. In addition, we found that CRBN was rapidly degraded upon exposure to PPARs agonists. These studies may help us fully elucidate the mechanisms through which PPARs affect CRBN expression level. Furthermore, our retrospective cohort study provided clinical evidence that the co-administration of PPAR agonists with IMiDs resulted in worse PFS and OS in patients with MM.

Our study is significant in several respects, providing the first evidence that PPARs agonists or antagonists may affect MM cell line sensitivity to lenalidomide by changing the methylation pattern of CpG islands in the CRBN promoter region. Moreover, PPAR agonists affect the outcomes of patients with MM when they are treated with IMiDs. This study further supports our previous research that PPAR agonists attenuate lenalidomide activity in MM and suggests that patients taking IMiDs avoid PPAR agonists. Our findings have important clinical relevance because over 1 in 4 myeloma patients has diabetes or dyslipidemia and many of these patients are taking PPAR agonists concurrently with IMiD treatment. Although additional and prospective studies are needed, our study suggested that for patients with myeloma and treated with IMiDs, the use of PPAR agonists should be avoided and other medications such as statins, metform, and insulin can be used for the management of dyslipidemia and/or type II diabetes.

A few studies are also consistent with our findings and support the association between CRBN methylation and IMiD resistance in MM. Haertle et al. identified a previously undescribed DNA hypermethylation in an active intronic CRBN enhancer [54]. Differential hypermethylation in this region was pronounced in IMiD-refractory MM. Methylation was significantly correlated with decreased CRBN expression levels. In vitro experiments with DNA methyltransferase inhibitors induced CRBN enhancer demethylation, and sensitizing effects of lenalidomide treatment were observed in two MM cell lines. However, other studies found no hypermethylation in the CRBN promoter in clinical samples [19]. Additionally, Dimopoulos et al. showed no involvement of promoter methylation in the regulation of CRBN expression in lenalidomide-resistant MM cell lines [55]. This inconsistency might be due to different treatment regimens used in the studies.

Apart from the prognostic value of CRBN level, CRBN genetic alterations such as gain/loss or mutation following lenalidomide treatment have been reported but the results were controversial. One study found mutations in CRBN in 6 (12%) out of 50 patients with relapsed MM [56], while another study found 3 patients with mutations in TP53 but no mutations in CRBN or IKZF1 out of 22 patients with relapsed MM [19]. Differences in treatment duration and combination with other agents could affect the incidence of mutations in CRBN following lenalidomide treatment. Further studies with larger patient sample size will be important to ascertain the incidence and role of mutations in CRBN and related genes in patients with relapsed MM.

Targeted protein degradation represents a rapidly growing area in drug discovery and development [57,58]. Moreover, small molecules or compounds that reduce targeted protein degradation, thereby increasing relative chemical sensitivity, also represent an important direction to overcome drug resistance. More recently, CRBN has gained popularity as an E3 ubiquitin ligase that can be hijacked by small-molecule degraders and used for targeted protein degradation [59,60]. The utility of CRBN in this context stems from the discovery that it binds IMiDs, resulting in the recruitment of IKZF1 and IKZF3 and their subsequent ubiquitination and proteasomal degradation [61]. Additionally, IMiDs can be incorporated into bifunctional degrader molecules (also called PROTACs), which can induce the degradation of target proteins beyond transcription factors, by bringing them into proximity to an E3 ligase [62,63,64]. These compounds are composed of an E3 ligase-recruiting element, such as IMiDs that recruit CRBN, and a binding motif for the target of interest. Bifunctional degraders function in the same manner as IMiDs, by facilitating the formation of ubiquitination-competent ternary complexes [65,66].

Drug-drug interactions between PPAR agonists and antitumor agents remain largely uncharacterized. Many anticancer treatment techniques interact with PPARγ ligands that synergistically enhance their efficiency. The combination of PPARγ ligands and tyrosine kinase inhibitors in chronic myelogenous leukemia is the most striking example [67]. Here, PPARγ activation renders leukemic stem cells that have previously been resistant to treatment susceptible to targeted therapy. In contrast, treatment with PPARγ ligands significantly alters the response and cell viability when co-treated with imatinib in chronic myeloid leukemia [67]. Therefore, given the possibility of interactions, a clinical evaluation of PPAR and the pharmacodynamic effects of co-administration with IMiDs will be useful to inform future clinical practices.

Aberrant lipid metabolism is recognized as a key feature of MM [68]. A previous study has reported a significant accumulation of lipids in MM cells after proteasome inhibition [69]. Our metabolic analysis, performed in a lenalidomide-resistant cell line, showed a similar trend after lenalidomide treatment in MM cell lines. This finding prompted us to hypothesize that MM cell survival depends on the maximal utilization of abnormally accumulated lipids. Whether lipid metabolism-modulating agents synergize with lenalidomide must be explored in the future. This requires additional research that is beyond the scope of this manuscript. With an in-depth understanding of tumor pathology and the evolution of new drug research and development technology, we believe that newer small-molecule compounds or partial agonists/antagonists will be developed in the near future.

## 5. Conclusions

To our knowledge, this is the first study to explore the response to IMiDs in patients who have MM with co-existing diabetes and/or dyslipidemia and treated with PPAR agonists. Between patients receiving IMiDs and patients receiving IMiDs and PPAR agonists, we found statistically significant differences in the overall response rate, PFS, and OS. More clinical samples should be used to detect the CRBN promoter methylation status. Furthermore, there were limitations to our study. It was a single center, retrospective analysis of data. Despite its limitations, the present study provides new information regarding drug-to-drug interaction that may be useful in guiding future studies.

## Figures and Tables

**Figure 1 cancers-14-05272-f001:**
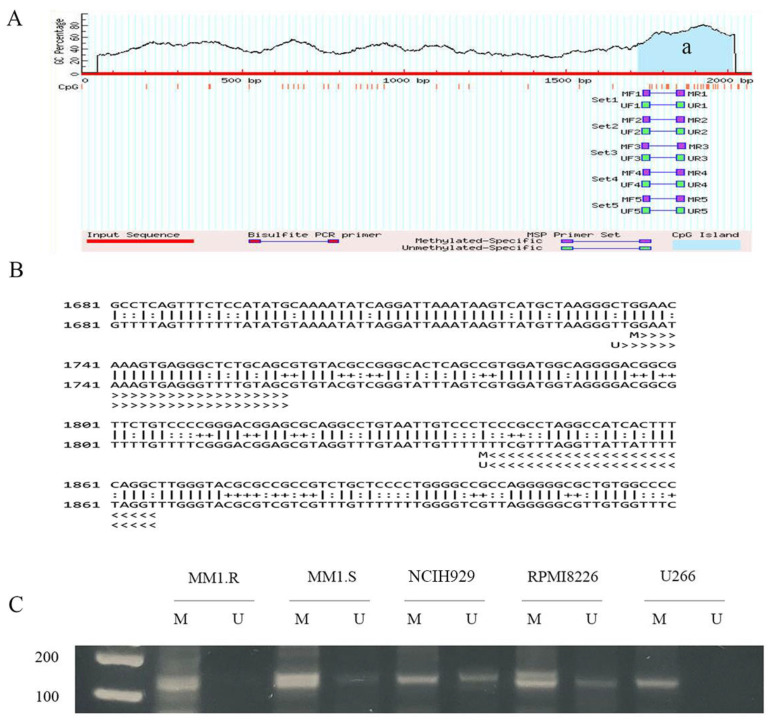
DNA methylation of CpG island in CRBN promoter region was associated with sensitivity to lenalidomide in MM cell lines. (**A**) CpG island prediction using MethPrimer software, ‘a’ represent the methylated region; (**B**) Predicted sequences of the CRBN design using MethPrimer. Methylated nucleotides are indicated with ‘+’, unmethylated nucleotides with ‘:’ and other nucleotides with ‘|’; (**C**) The productions from the methylation specific PCR sun on agarose gels. Row key: M = methylation specific reaction, U = unmethylated specific reaction. The uncropped blots are shown in Appendix A.

**Figure 2 cancers-14-05272-f002:**
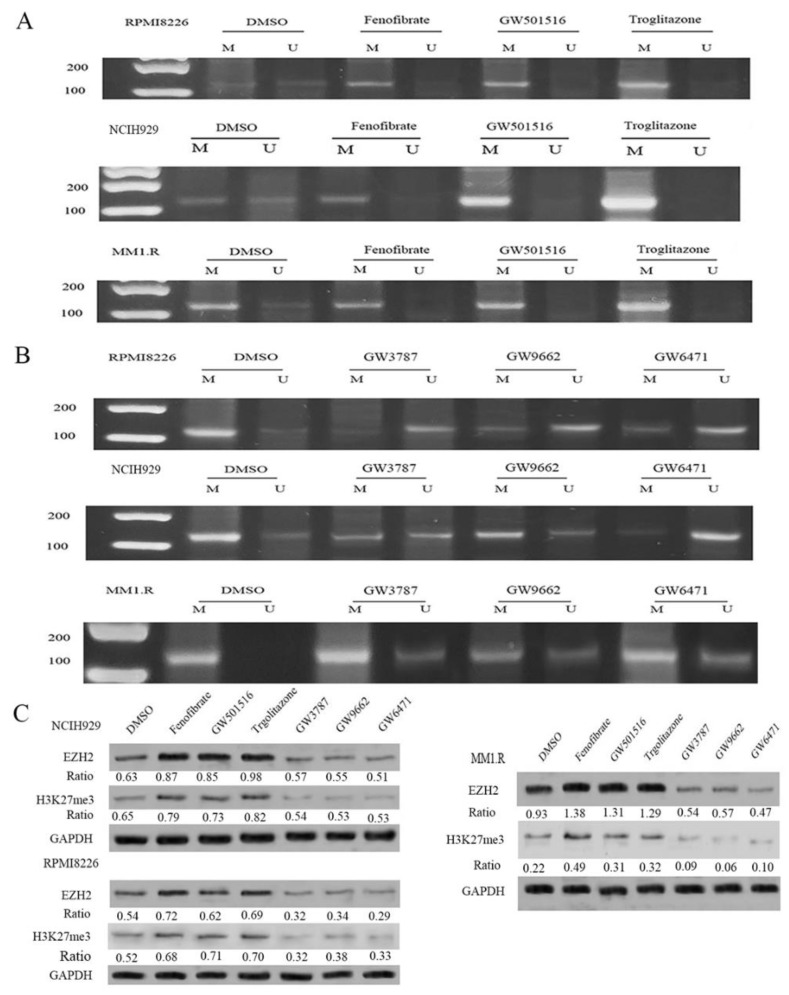
PPAR agonists or antagonists affect the methylation pattern of CpG island in CRBN promoter region. (**A**,**B**) RPMI8226, NCIH929 and MM1.R cells were treated with 5µM PPAR agonists (Fenofibrate, GW501516, and Troglitazone) or 10 µM PPAR antagonists (GW3787, GW9662, and GW6471) 48 h. MSP shows the different pattern of methylation status. M: results of methylation specific PCR amplification. U: results of non-methylation specific PCR amplification; (**C**) RPMI8226, NCIH929 and MM1.R cells were treated with 5 µM PPAR agonists (Fenofibrate, GW501516, and Troglitazone) or 10 µM PPAR antagonists (GW3787, GW9662, and GW6471) 48 h. EZH2 and H3K27me3 expression were measured by Western blot. The intensity of the level of EZH2, H3K27m23 and GAPDH was quantitated and the ratio of EZH2/H3K27m3 vs. GAPDH was indicatd. The uncropped blots are shown in Appendix A.

**Figure 3 cancers-14-05272-f003:**
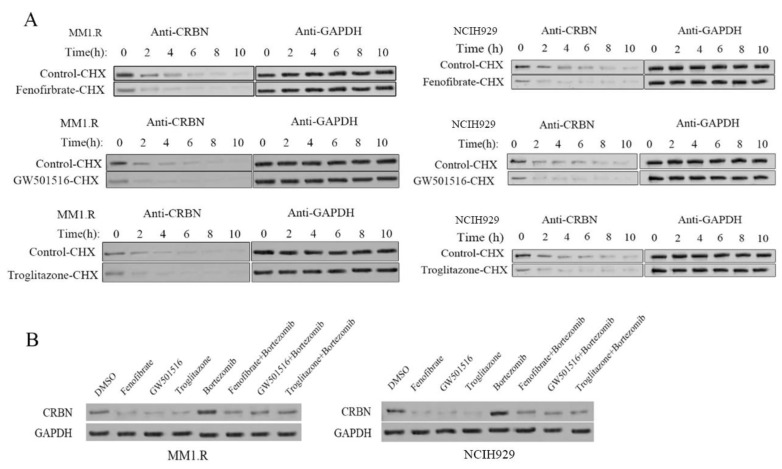
CRBN is rapidly degraded in the presence of PPAR agonists. (**A**) MM1.R and NCIH929 cells were treated with 5 µM Fenofibrate, GW501516 and Troglitazone. The protein degradation of CRBN was assessed following treatment with 100 µg/mL cycloheximide (CHX) for 0–10 h; (**B**) Proteasome inhibitor (bortezomib) prevented the degradation of CRBN by PPAR agonists. MM1.R and NCIH929 cells were treated with DMSO control, bortezomib (50 nM) for 1 h, followed by treated with DMSO or 5 µM Fenofibrate, GW501516 and Troglitazone for an additional 6 h. Whole cell lysate was prepared and analyzed for CRBN expression by Western blot analysis. Data were representative of 2 separate sets of experiments. The uncropped blots are shown in Appendix A.

**Figure 4 cancers-14-05272-f004:**
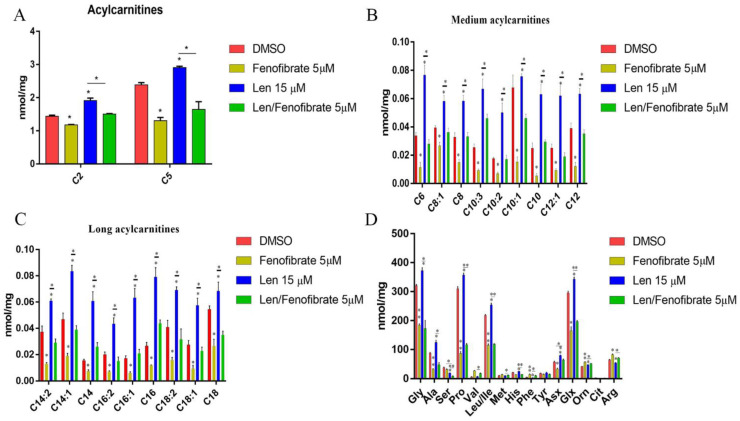
Fenofibrate and lenalidomide has opposite effects of lipid and amino acid metabolism. (**A**). MM1.R was treated with DMSO control, 5 µM Fenofibrate, 15 µM lenalidomide, and a combination group 48 h. Collect samples and subject to the Duke Metabolic Core Facility. C2, C5 Acylcarnitines (**A**), medium-chain acylcarnitines (**B**), long-chain acylcarnitines (**C**) and several amino acids (**D**) were analyzed. *: *p* < 0.05; **: *p* < 0.01.

**Figure 5 cancers-14-05272-f005:**
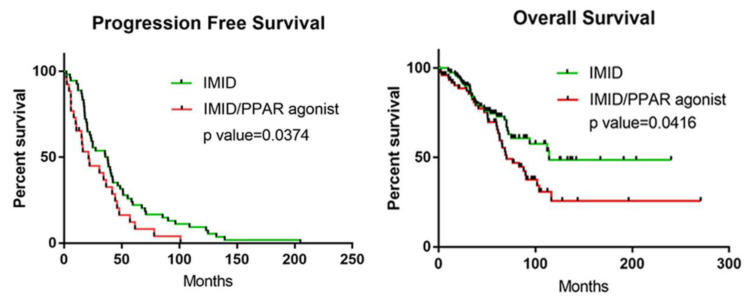
Coadministration of PPAR agonists with IMiDs is associated with worse progression-free survival and overall survival. **Left** panel: progression-free survival. **Right** panel: overall survival.

## Data Availability

The data presented in this study are available in this article (and Appendix A).

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
