# Peer review of "Mechanistic Studies and a Retrospective Cohort Study: The Interaction between PPAR Agonists and Immunomodulatory Agents in Multiple Myeloma"

_cancers, 2022, doi:10.3390/cancers14215272_

Round 1

Reviewer 1 Report

In this paper, the authors show that PPAR agonists which are given for the treatment of dyslipidemia and diabetes mellitus may attenuate the action of IMiDs in the treatment of MM through methylation of CRBNs in vitro. Furthermore, they showed the prognosis of MM patients treated with PPAR agonists in a clinical cohort study. Since the reviewer believes that this finding will be very beneficial for the future treatment of MM, it would be worthy of publication in this journal. The following improvements would be requested for publication.

1. Figure 1- is the H929 on the label a misstatement of NCIH929? If so, please correct it.

2. Figure 2B- MM.1R does not appear to be demethylated by the PPAR antagonist. Please do a follow-up experiment.

3. Fig. 2C- It is not very clear that the induction of H3k27me3 expression by PPAR agonisit in H929 and RPMI8226. Please perform a follow-up experiment or quantification by densitometry.

4. Do metformin, insulin and statins affect methylation of CRBN? Wouldn't this be important data if it shows a difference from PPAR agonists?

Author Response

We greatly appreciate the time and effort the reviewers spent reviewing our manuscript and we thank them for their constructive comments and excellent critiques.  The manuscript was revised accordingly.  The changes were tracked in the revised manuscript. The point-to-point responses to the Reviewers’ comments are included below.

Point-to-point responses to reviewer's comments:

1.  Figure 1- is the H929 on the label a misstatement of NCIH929? If so, please correct it.

Response: Yes. The "H929" on the label is "NCIH929". We have corrected it in Figure 1 in the revised manuscript.

2. Figure 2B- MM.1R does not appear to be demethylated by the PPAR antagonist. Please do a follow-up experiment.

Response:  We have repeated the experiment and the data are now included in Figure 2B of the revised manuscript. The data showed clear demethylation by the PPAR antagonist in MM.1R cells

3. Fig. 2C- It is not very clear that the induction of H3k27me3 expression by PPAR agonisit in H929 and RPMI8226. Please perform a follow-up experiment or quantification by densitometry.

Response: We have repeated the experiments and the results showed clear induction of H3k27me3 expression by PPAR agonists in NCIH929 and RPMI8226. The data are now included in Figure 2C of the revised manuscript.

4. Do metformin, insulin and statins affect methylation of CRBN? Wouldn't this be important data if it shows a difference from PPAR agonists?

Response: This is an excellent question but we do not have the definitive answers. We plan to do these experiments in the near future and will report the findings in a separate manuscript.

Reviewer 2 Report

The paper by Wu et al reports on the results of a retrospective analysis of the effects of PPAR agonist on simultaneous treatment of MM patients with lenalidomide (lena). The authors were able to confirm that the role of PPAR agonist is to downregulate cereblon (CRBN) which in turn reduces the action of lena.

The authors applied with competence sensitive techniques to follow the effects on methylation on CRBN and the resulting effects on the functionality of the system. The proof-of-principle of their study in vitro and using myeloma cell lines resistant (or non) to lena were confirmed by an original approach based on a retrospective analysis of a sample of MM patients treated or not with PPAR agonists. The PFS and OS parameters analysed in the patients samples confirmed that the use of PPAR agonists is paralleled by worse general conditions.

In my view, this is an original and accurate evaluation of a project of basic science with potential follow-out in clinical practice.

Author Response

We greatly appreciate the time and effort the reviewer spent reviewing our manuscript and we thank you for your comment that "this is an original and accurate evaluation of a project of basic science with potential follow-out in clinical practice".